# Presynaptic inhibition of dopamine neurons controls optimistic bias

**Nobuhiro Yamagata[1]\*, Takahiro Ezaki[2], Takahiro Takahashi[1], Hongyang Wu[1], Hiromu Tanimoto[1]\***

[1]Graduate School of Life Sciences, Tohoku University, Sendai, Japan; [2]Research Center for Advanced Science and Technology, The University of Tokyo, Tokyo, Japan

**Abstract** Regulation of reward signaling in the brain is critical for appropriate judgement of the environment and self. In *Drosophila*, the protocerebral anterior medial (PAM) cluster dopamine neurons mediate reward signals. Here, we show that localized inhibitory input to the presynaptic terminals of the PAM neurons titrates olfactory reward memory and controls memory specificity. The inhibitory regulation was mediated by metabotropic gamma-aminobutyric acid (GABA) receptors clustered in presynaptic microdomain of the PAM boutons. Cell type-specific silencing the GABA receptors enhanced memory by augmenting internal reward signals. Strikingly, the disruption of GABA signaling reduced memory specificity to the rewarded odor by changing local odor representations in the presynaptic terminals of the PAM neurons. The inhibitory microcircuit of the dopamine neurons is thus crucial for both reward values and memory specificity. Maladaptive presynaptic regulation causes optimistic cognitive bias.

**\*For correspondence:**
nobuhiro.yamagata.a5@tohoku.ac.jp (NY);
hiromut@m.tohoku.ac.jp (HT)

**Competing interests:** The authors declare that no competing interests exist.

## Introduction

Regulation of reward signaling in the brain is critical for maximizing positive outcomes and for avoiding futile costs of the behaviors at the same time. Across animal phyla, dopamine neurons are primarily involved in reward processing (*Brembs et al., 2002*; *Tobler et al., 2005*; *Liu et al., 2012*; *Ichinose et al., 2017*). In the fruit fly *Drosophila melanogaster*, a subset of dopamine neurons in the protocerebral anterior medial (PAM) cluster mediates the reinforcement property of sugar reward (*Burke et al., 2012*; *Liu et al., 2012*). In olfactory learning, dopamine input to the mushroom body (MB) causes changes in preference of a simultaneously presented odor by modulating the output of odor-representing MB intrinsic neurons, Kenyon cells (KCs) (*Séjourné et al., 2011*; *Boto et al., 2014*; *Cohn et al., 2015*; *Owald et al., 2015*; *Louis et al., 2018*; *Hige et al., 2015*; *Bilz et al., 2020*). Such associative presentations of odor and electric shocks were reported to change the activity of MB-projecting dopamine neurons (*Riemensperger et al., 2005*). Recent studies (*Hattori et al., 2017*; *Cervantes-Sandoval et al., 2017*; *Takemura et al., 2017*) suggest that axon terminals of the dopamine neurons locally integrate olfactory inputs to function as multiple independent units, though such subcellular reward processing has yet to be examined.

## Results and discussion

To understand neuronal mechanisms for the regulation of reward processing, we here focused on gamma-aminobutyric acid (GABA) signaling in the PAM neurons. Six GABA receptor genes are identified in the fly genome. We silenced the expression of each receptor gene in the PAM cluster neurons by targeting transgenic RNAi (*Ni et al., 2011*) and tested their appetitive olfactory memory (*Figure 1A*). We found increased memory performance by downregulation of a metabotropic GABA receptor, *GABA-B-R3* (*Figure 1A* and *Figure 1—figure supplement 1A-B*).

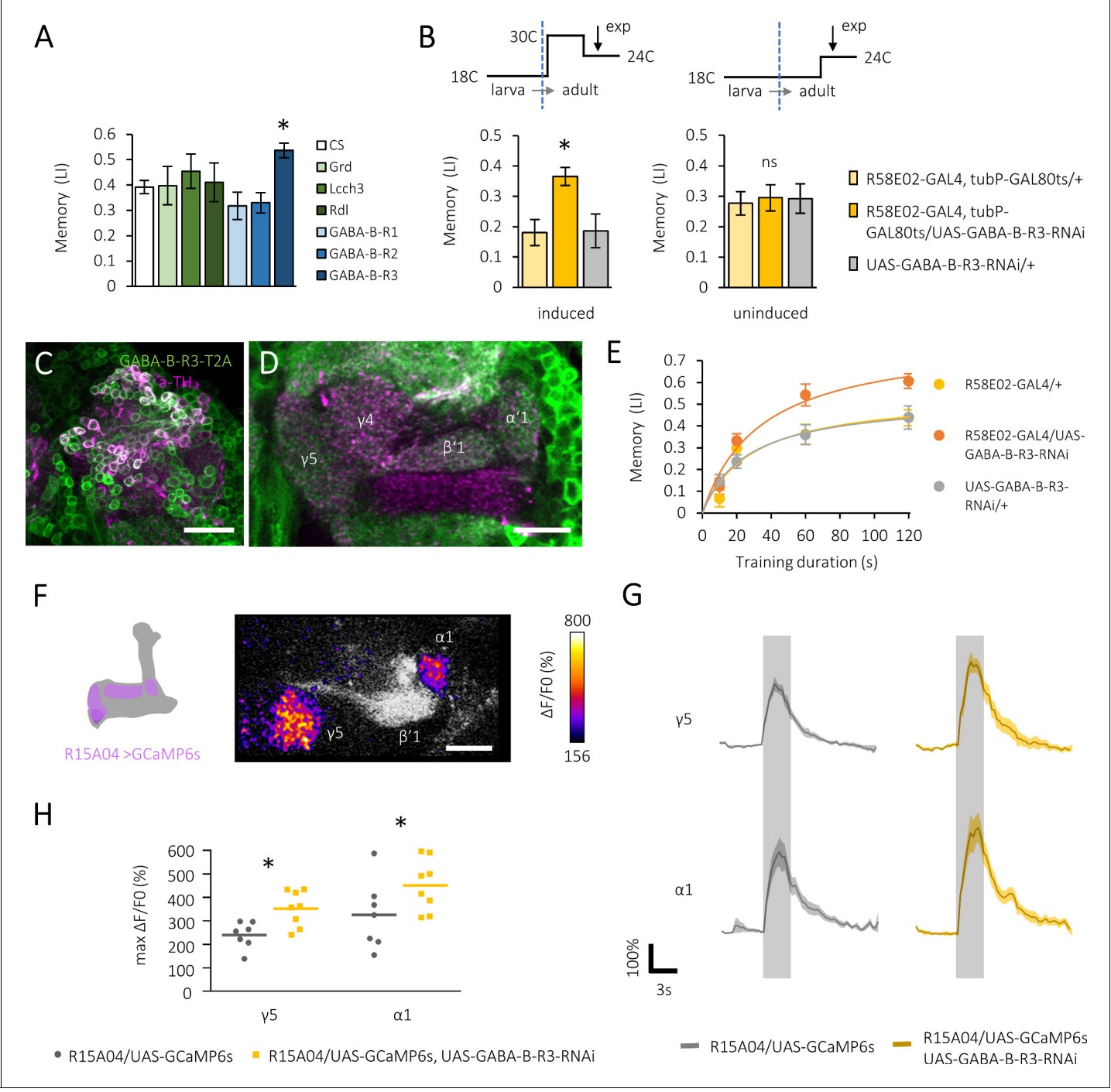

**Figure 1.** Gamma-aminobutyric acid (GABA)-B-R3 suppresses sugar reward. (**A**) Cell type-specific GABA receptor silencing in the protocerebral anterior medial (PAM) neurons directed by *R58E02-GAL4*, showing *GABA-B-R3*-specific memory increment. *q < 0.008 (Benjamini and Hochberg method, N = 8, 23). Mean ± SEM are shown hereafter. (**B**) Acute silencing of *GABA-B-R3* in the PAM neurons using *Tub-GAL80^ts* and heat-induced inactivation. *p<0.05 (Sidak's test, N = 10, 12; Dunn's multiple comparisons test, N = 7, 8). (**C, D**) A substack projection image and a single optical slice showing endogenous *GABA-B-R3* expression in somata of PAM neurons (**C**) and their axon terminal profiles in the mushroom body (MB) (**D**) by using *GABA-B-R3-T2A-GAL4*. Anti-tyrosine hydroxylase (TH) antibody signal (magenta) for labeling PAM neurons. Scale bars, 100 µm (**C**), 20 µm (**D**). (**E**) Memory acquisition curves of knock-down and control flies. Hyperbola curve fitting and subsequent permutation tests (*Figure 1—figure supplement 1E–F*) reveal an altered plateau level but not the acquisition speed in knock-down flies. N = 12. (**F**) Superimposed and color-coded sugar-evoked calcium signal (ΔF/F_0) in a subset of PAM neurons measured in *R15A04-GAL4/UAS-GCaMP6s, UAS-mCD8::RFP* flies. (**G**) Time course of the calcium transients in defined compartments of the MB in control (left) and *GABA-B-R3* knock-down (right) flies. Gray shades indicate sugar stimulation for 3 s. Mean ± SEM, N = 7, 8. (**H**) *GABA-B-R3* knock-down significantly increases sugar-evoked peak calcium transients in γ5 and α1 neurons. *p<0.05 (Holm-Sidak's test, N = 7, 8).

Figure 1 continued

The online version of this article includes the following source data and figure supplement(s) for figure 1:

**Source data 1.** GABA receptor silencing in the PAM neurons and the effects on appetitive memory.
**Source data 2.** Acute suppression of GABA-B-R3.
**Source data 3.** Memory acquisition curve in control and GABA-B-R3 knock-down flies.
**Source data 4.** Sugar response in the PAM neurons in control and GABA-B-R3 knock-down flies.
**Figure supplement 1.** Gamma-aminobutyric acid (GABA)-B-R3 knock-down in the protocerebral anterior medial (PAM) neurons increases memory strength.
**Figure supplement 1—source data 1.** Appetitive memory score in control and GABA-B-R3 knock-down flies.
**Figure supplement 1—source data 2.** Appetitive memory score in control and GABA-B-R3 knock-down flies with a second RNAi line.
**Figure supplement 1—source data 3.** Memory acquisition curve in wild-type flies.

We next examined endogenous *GABA-B-R3* expression in the adult brain using the intronic CRISPR-Mediated Integration Cassette (CRIMIC) insertion of *T2A-GAL4* (*Lee et al., 2018*). The T2A self-cleaving peptide between the target protein and GAL4 allows bi-cistronic translation by a ribosome skipping mechanism (*Diao and White, 2012*). *GABA-B-R3* was expressed broadly in the brain, including the majority of the PAM cluster neurons (*Figure 1C–D*), whereas the expression was weak in KCs (*Figure 1D* and *Figure 1—figure supplement 1C*). There was no notable morphological alteration in the brain of knock-down flies (data not shown). Consistently, adult stage-specific *GABA-B-R3* silencing in the PAM neurons using *Tub-GAL80^{ts}* (*McGuire et al., 2003*) similarly enhanced appetitive memory performance (*Figure 1B*). Without transgene induction, their appetitive memory was indistinguishable from the controls (*Figure 1B*).

Increased learning speed and/or performance plateau may underlie the enhanced appetitive memory in the *GABA-B-R3* knock-down flies. We attempted to distinguish these possibilities by characterizing their memory acquisition (*Figure 1E* and *Figure 1—figure supplement 1D*). The performance of *R58E02-GAL4/UAS-GABA-B-R3-RNAi* flies reached a significantly higher asymptote than control genotypes without changing the acquisition speed (*Figure 1E* and *Figure 1—figure supplement 1E–F*). In a learning theory, the magnitude of reinforcement is the determinant for the plateau of the acquisition curve (*Rescorla, 1972*), suggesting that sugar reward was perceived more strongly with enhanced dopaminergic activity in the *GABA-B-R3* knock-down flies. Live calcium imaging at terminal branches of the reward-related PAM neurons (i.e., PAM-γ5 and -α1) revealed the augmented sugar responses upon downregulating *GABA-B-R3* (*Figure 1F–H*). We thus conclude that GABA-B-R3 signaling is required for negative regulation of the sugar reward.

We visualized the localization of GABA-B-R3 proteins using a GFP-tagged reporter (*Sarov et al., 2016*). GABA-B-R3 proteins were heavily localized to the presynaptic terminals of the PAM neurons (*Figure 2A–B* and *Figure 2—figure supplement 1A*). We thus hypothesized that presynaptic inhibition of dopamine neurons within the MB controls the gain of reward signals. A single pair of the GABAergic anterior paired lateral (APL) neurons was reported to massively innervate the entire MB and to be involved in olfactory learning (*Liu and Davis, 2009*). Differential labeling of the PAM and APL neurons revealed that their ramifications abut on each other (*Figure 2C–D*). Consistently, we found enhanced reward memory in knock-down flies for *glutamic acid decarboxylase 1* (*Gad1*) and *vesicular GABA transporter* (*VGAT*) in the APL neuron (*Figure 2E*). This result not only underscores the importance of GABA metabolism in the APL neurons, but suggests the role of the inhibitory microcircuit in the MB for the gain control of the reward value. We therefore examined the local inhibition hypothesis by comparing sugar responses in the dendrites and presynaptic terminals of the PAM neurons (*Figure 2F* and *Figure 2—figure supplement 1B–C*). The enhanced calcium activity upon *GABA-B-R3* knock-down was much more pronounced in the presynaptic terminals (*Figure 2G–H*). Therefore, GABAergic signals from the APL neurons negatively control the reward gain at the output site of the PAM neurons through GABA-B-R3 signaling in the MB.

To quantify the local activity regulation in the PAM terminals, we measured calcium influx at active zones using the ratiometric calcium sensor Brp::GCaMP6s::mCherry (*Kiragasi et al., 2017*). This sensor is composed of GCaMP fused to calcium insensitive mCherry and targeted to active zones using the short fragment of Brp, enabling the measurement of local calcium influx at active zones (*Kiragasi et al., 2017*). Immunolabelling confirmed the localization of the sensor

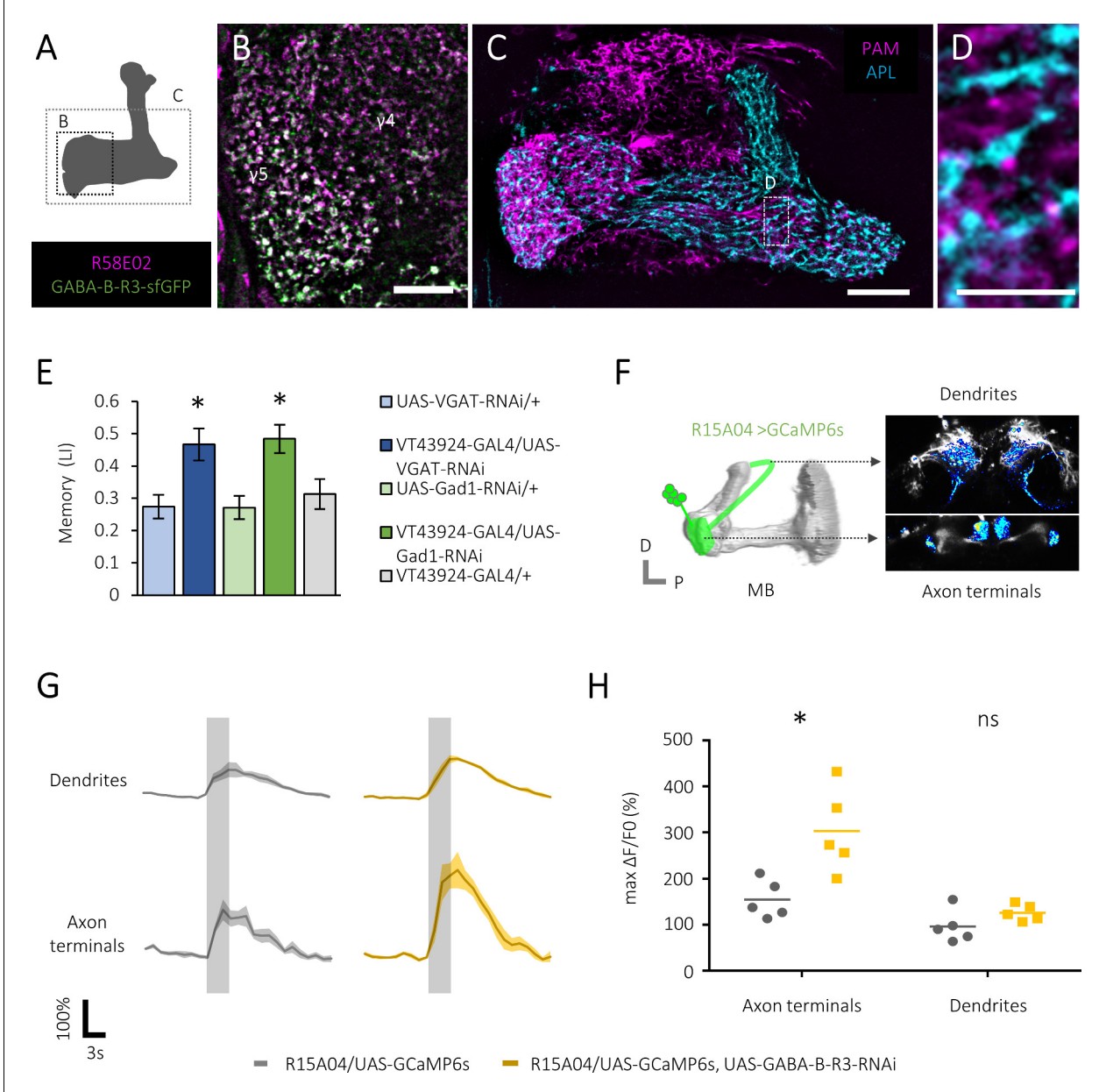

**Figure 2.** Presynaptic GABAergic inhibition of the protocerebral anterior medial (PAM) neurons. (**A**) A schematic showing brain regions visualized in subsequent figures. (**B**) A substack projection image of double labeling of GFP-tagged GABA-B-R3 protein and PAM neurons using *GABA-B-R3-sfGFP-TVPTBF, R58E02-GAL4/UAS-mCD8::RFP* fly. Presynaptic localization of GABA-B-R3 protein in the γ5 and α1 (*Figure 2—figure supplement 1A*) compartments of the mushroom body (MB). Scale bar, 10 μm. (**C–D**) A substack projection image showing double labeling of the PAM and anterior paired lateral (APL) neurons visualized by *VT43924-GAL4/UAS-mCD8::RFP, R58E02-LexA/LexAop-rCD2::GFP*. These axonal profiles co-localize at close proximity in the MB. A single optical slice of the inset in C is magnified in D. Scale bars, 10 μm (**C**), 2 μm (**D**). (**E**) Increased appetitive memory score by shRNA-mediated silencing of GABA neurotransmission-related genes in the APL neurons. *p<0.05 (Sidak's test, N = 11–13). (**F**) A cartoon depicting volumetric imaging from axon terminals and dendritic fields of PAM neurons using a z-objective piezo actuator. (**G**) Time course of calcium transients (ΔF/F$_0$) near-simultaneously recorded from axon terminals and dendrites of PAM-γ5 neurons (*Figure 2—figure supplement 1*) of control (left) and *GABA-B-R3* knock-down (right) flies. Gray shades indicate sugar stimulation for 3 s. Mean ± SEM, N = 5. (**H**) Significantly increased sugar-evoked peak calcium transients of *GABA-B-R3* knock-down flies in axon terminals. *p<0.05 (Dunn's test, N = 6, 7).

The online version of this article includes the following source data and figure supplement(s) for figure 2:

**Source data 1.** Inhibition of GABA neurotransmission in APL neurons.

**Source data 2.** Sugar response in the axon terminals and dendrites of the PAM neurons.

**Figure supplement 1.** Presynaptic gamma-aminobutyric acid (GABA)-B-R3 localization and volumetric Ca$^{2+}$ imaging from a defined protocerebral anterior medial (PAM) cell class.

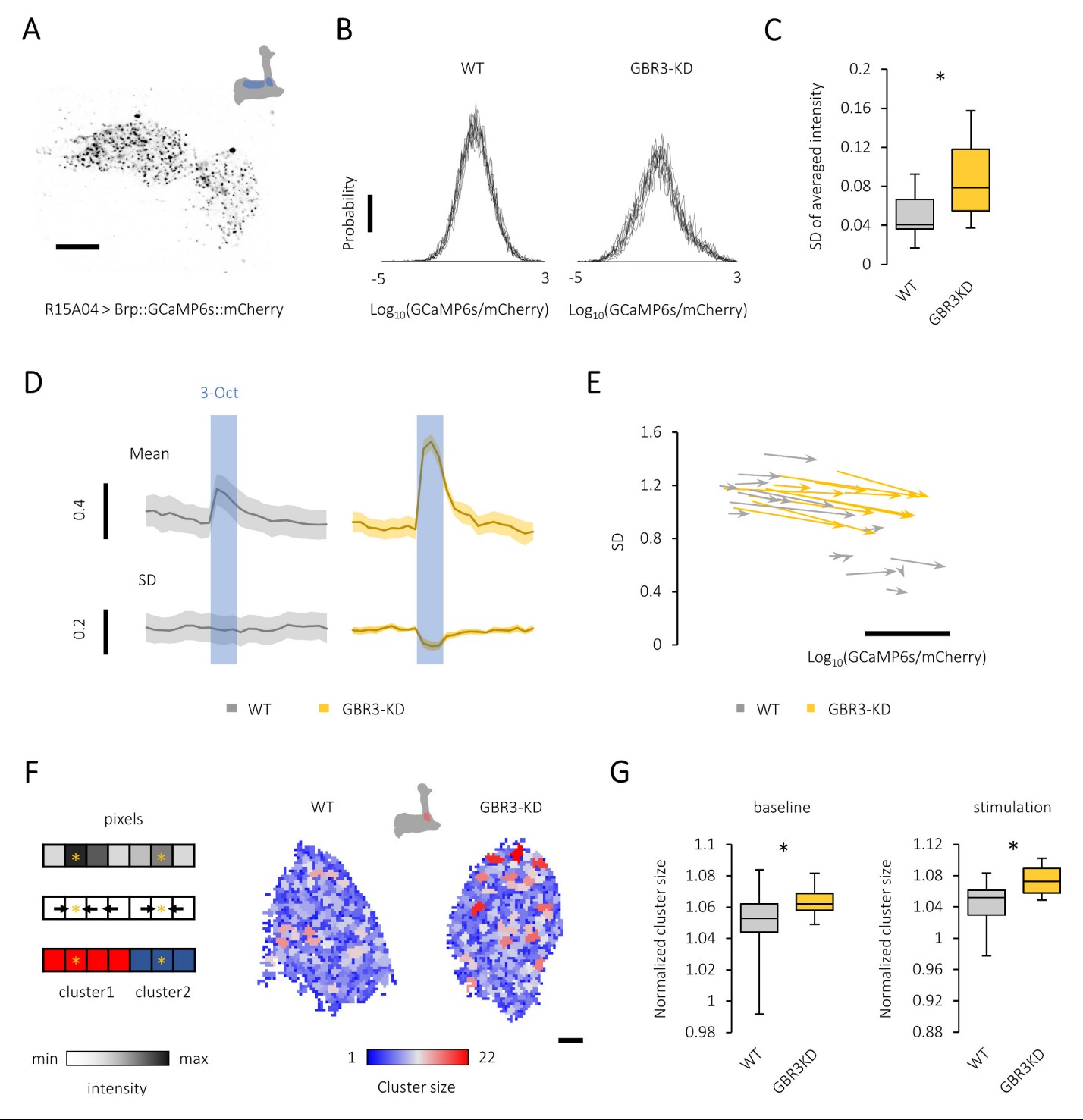

**Figure 3.** Gamma-aminobutyric acid (GABA)-B-R3 controls presynaptic activity patterns in the protocerebral anterior medial (PAM) neurons. (**A**) A single optical slice showing punctate sensor localization in the mushroom body (MB) lobe of *R15A04-GAL4/UAS-Brp::GCaMP6s::mCherry* flies. Scale bar, 10 μm. (**B**) The histograms of GFP/RFP signals from nine different time points (PAM-α1 terminals) reveal a large calcium diversity among active zones and the temporal instability in the knock-down fly (right). The probability distribution of the active-zone calcium mostly followed the Gaussian distribution, characteristics of which are well represented by the mean and standard deviation (SD). Scale bar, 0.01. (**C**) The temporal variance of spatially averaged intensities is larger in the *GABA-B-R3* knock-down PAM neurons. *p<0.001 (Mann-Whitney test, N = 24, 17). (**D**) Time course of odor-evoked calcium transient (mean) and the variance (SD) of PAM-α1 terminals in control (left) and GABA-B-R3 knock-down (right) flies. The blue shade indicates odor stimulation for 3 s. Mean ± SEM, N = 6–8. (**E**) Map of the probability distribution of active-zone calcium intensity in the PAM-α1 terminals for each

*Figure 3 continued on next page*

*Figure 3 continued*

individual. The abscissa and ordinate represent the mean and SD of the signal distribution, respectively. The clear inverse correlation between the mean and the variance in wild-type terminals (r = −0.82, p<10⁻⁶) may represent individually defined unique set points of activity levels. This structured individual difference is disrupted by *GABA-B-R3* RNAi (r = 0.53, p=0.03). The start and end of an arrow represent the probability distribution of active-zone calcium before and during the odor stimulation, respectively. Scale bar, 0.5 in Log$_{10}$(GCaMP6s/mCherry). (F) Local peaks in the PAM terminals are color-coded for their cluster sizes (right). Schematic example of the algorithm finding the cluster structure applied to a one-dimensional system (Left). Spatial distribution of calcium intensity (Left upper). Finding steepest paths from each pixel to local peaks by computing the gradient (i.e., the difference of the intensity values between neighboring pixels)(Left middle). Note that in two-dimensional systems we analyzed, each pixel had four neighboring pixels. Clustering based on the paths to local peaks (Left lower). Pixels having the same destination (i.e., local peak) are clustered together. (G) The average area per peak is significantly larger in the *GABA-B-R3* knock-down flies during baseline activity (*p<0.05, Mann-Whitney test, N = 22, 17) and odor stimulation (*p<0.05, t-test, N = 15, 11).

The online version of this article includes the following source data and figure supplement(s) for figure 3:

**Source data 1.** Temporal variance of spatially averaged intensities in control and GABA-B-R3 knock-down flies.
**Source data 2.** Probability distribution of active-zone calcium intensity before and during odor stimulation.
**Source data 3.** Average cluster size in control and GABA-B-R3 knock-down flies.
**Figure supplement 1.** Spatial and temporal characteristics of active-zone calcium signal in the presynapses of protocerebral anterior medial (PAM) neurons.
**Figure supplement 1—source data 1.** Standard deviation of the active-zone calcium intensity.
**Figure supplement 1—source data 2.** obability distribution of active-zone calcium intensity before and during sugar stimulation.
**Figure supplement 1—source data 3.** Normalized peak size in control and GABA-B-R3 knock-down flies.

proteins to active zones of the PAM neurons in the MB (*Figure 3A* and *Figure 3—figure supplement 1A*).

As the basal activities of dopamine neurons represent physiological information (*Ichinose et al., 2017*), we characterized the regulation of active zone calcium in the PAM terminals without overt stimulation. Live imaging of punctate Brp::GCaMP6s::mCherry signals in the PAM terminals visualized a large heterogeneity of calcium levels among active zones (*Figure 3B*, *Figure 3—figure supplement 1B* and *Figure 3—figure supplement 1C*). This heterogeneity tended to increase upon silencing *GABA-B-R3* (*Figure 3—figure supplement 1D*). Furthermore, we found that Brp::GCaMP6s::mCherry signals had temporal fluctuations, which was amplified by silencing *GABA-B-R3* (*Figure 3B* and *Figure 3C*). GABA-B-R3 in the PAM terminals may thus stabilize the basal presynaptic activity. This suggests that GABA inhibition contributes to the robustness of activity against local perturbations.

System robustness is often related to refined regulation of activity patterns by which information is efficiently coded (*Hesse and Gross, 2014*). As active-zone calcium in the PAM terminals is likely to reflect local input in the MB (see *Figure 2G and H*), we hypothesized that GABA-B-R3 controls the spatial representation of odor information in the PAM terminals. In control flies, odor presentations barely changed the distribution of active-zone calcium with a marginal increase of the overall signal intensity (*Figure 3D–E* and *Figure 3—figure supplement 1C*). In contrast, sugar ingestion induced global presynaptic calcium increase and decreased the signal heterogeneity (*Figure 3—figure supplement 1D*). These stimulus-specific presynaptic responses are likely to reflect differential input sites of odor and sugar signals, that is, pre- and post-synaptic sites of the PAM neurons, respectively. Strikingly, *GABA-B-R3* knock-down flies responded to an olfactory stimulation much more strongly (*Figure 3D–E*), as if in response to a global sugar stimulation (*Figure 3—figure supplement 1D*). These results suggest that the local control of PAM presynaptic activity by GABA-B-R3 regulates the odor responses, possibly by refining its spatial representation.

To further characterize how GABA-B-R3 regulates the structure of presynaptic activity in PAM terminals, we examined the patterns of local peaks of Brp::GCaMP6s::mCherry signals by introducing a spatial measure. We found that the average peak area (i.e., cluster size) became larger upon knocking down *GABA-B-R3* both before and during the odor stimulation (*Figure 3F–G*). Note that we did not observe a clear correlation between the cluster size and intensity of pixels in the cluster (data not shown), suggesting that the cluster size measure provides information orthogonal to the intensity value of calcium signals. We also quantified the size of high-intensity pixels using another measure, which we call peak size (see Materials and methods; *Figure 3—figure supplement 1F*). We further found that intense Brp::GCaMP6s::mCherry signals are more spatially clustered in the knock-

down PAM terminals, further corroborating the fine spatial regulation of calcium signals by GABA-B-R3. Since KC and PAM terminals form mutual synapses in the MB lobe (*Takemura et al., 2017*; *Cervantes-Sandoval et al., 2017*), the overall disinhibition in the knock-down terminals may impair the selective delivery of dopaminergic reward signals to odor-activated KCs.

To test if presynaptic GABA-B-R3 signaling controls odor representations, we examined the specificity of memory using odor generalization. Conditioned odor approach of wild-type flies drops by increasing the blend ratio of a contaminant odor to the trained odor (*Ichinose et al., 2015*; *Chen et al., 2017*). Strikingly, the downregulation of *GABA-B-R3* in the PAM neurons resulted in a broader generalization profile, indicating that reward memory in the knock-down flies is less specific to the learned odor (*Figure 4A–B*). Moreover, we found similarly broadened generalization profiles

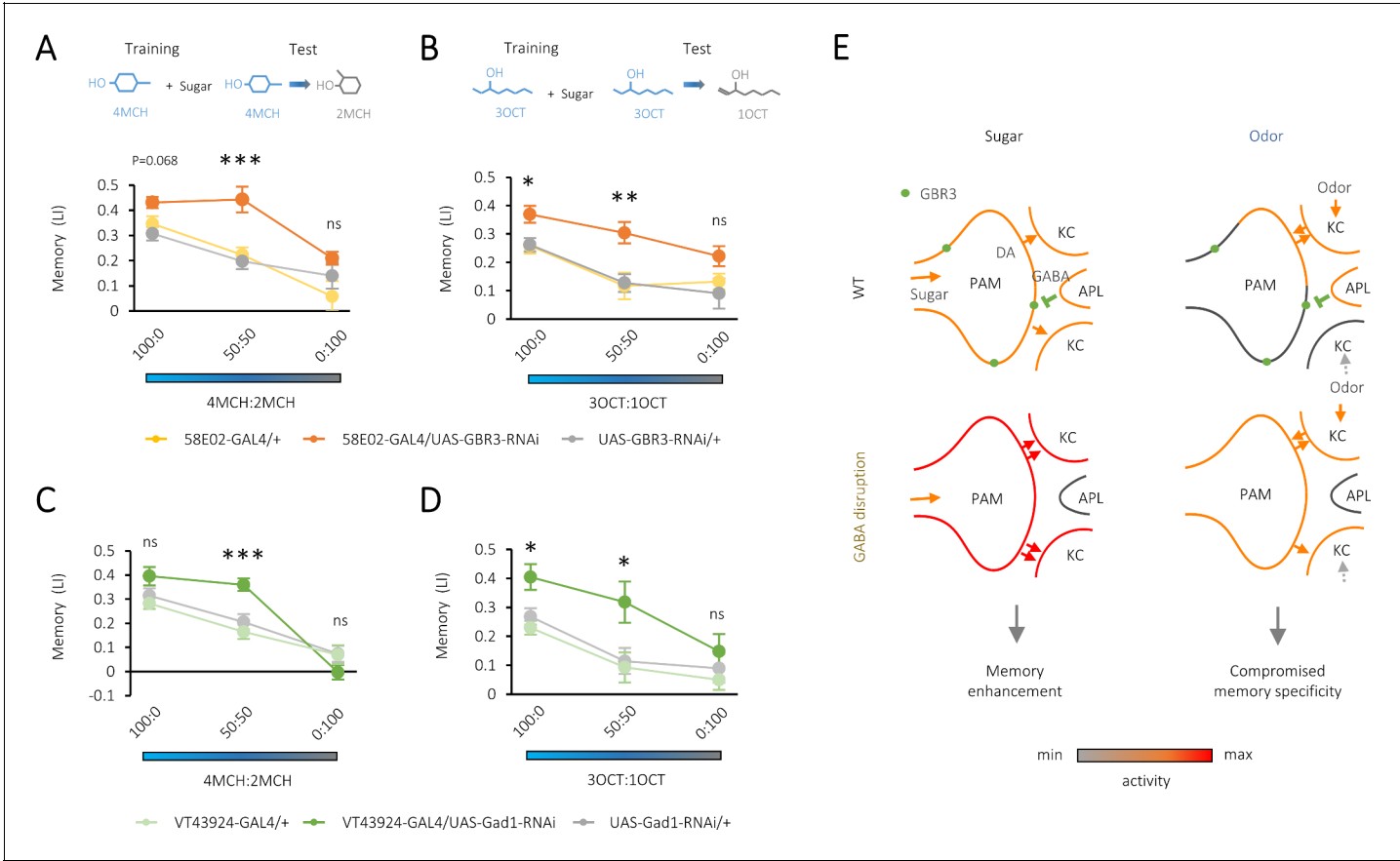

**Figure 4.** Compromised odor specificity in the rewarded memory by attenuating local gamma-aminobutyric acid (GABA) signaling in the mushroom body. (A–D) Olfactory generalization of appetitive memory depends on the blend ratio of a contaminant to a trained odor 4-methylcyclohexanol (4MCH) (A, C) or octan-3-ol (3OCT) (B, D). Significantly broader generalization in *R58E02-GAL4/UAS-GABA-B-R3-RNAi* flies (A, B) and *VT43924-GAL4/UAS-Gad1-RNAi* flies (C, D) compared to the respective parental controls. *p<0.05; ***p<0.001 (Sidak's test across genotypes at each blend ratio, N = 8–20 [A, B], N = 9–12 [C, D]). (E) A model for dual behavioral roles of GABA-B-R3 in the protocerebral anterior medial (PAM) terminal in the mushroom body (MB). GABA-B-R3 controls the overall gain (left) and localization (right) of sugar- and odor-evoked terminal activity of PAM neurons, respectively. Note that input sources of sugar and odor response are different. In an odor response, reciprocal PAM-KC (Kenyon cell) synapses serve as KC activity-dependent local enhancers for reward signaling from the PAM synapses. In wild-type flies, this local modulation is restricted to the PAM synapses onto the odor-activated KCs, which laterally inhibit activities of surrounding ones via KC-APL (anterior paired lateral) feedback. GABA-B-R3 knock-down in the PAM neurons eliminates this negative feedback from the APL, reducing the activity contrast within presynaptic terminals of the PAM neurons and the selectivity of memory to the rewarded odor.

The online version of this article includes the following source data for figure 4:

**Source data 1.** Olfactory generalization of appetitive memory in control and GABA-B-R3 knock-down flies (4MCH).
**Source data 2.** Olfactory generalization of appetitive memory in control and GABA-B-R3 knock-down flies (3OCT).
**Source data 3.** Olfactory generalization of appetitive memory in control and Gad1 knock-down flies (4MCH).
**Source data 4.** Olfactory generalization of appetitive memory in control and Gad1 knock-down flies (3OCT).

by downregulating GAD1 expression in the APL neurons (*Figure 4C–D*). We altogether conclude that local GABAergic inhibition of the PAM neurons regulates the intensity and specificity of reward memory (*Figure 4E*).

Our results indicate that presynaptic modulation of the PAM neurons is a critical component for determining the magnitude of dopaminergic reward signals. Notably, abolition of the local GABAergic input to the PAM terminals not only enhanced the internal reward intensity but compromised memory specificity (*Figures 1* and *4*). These behavioral alterations can be explained by a dual physiological role of GABA-B-R3, that is, the gain control and the spatial segmentation of dopaminergic reward signals in the PAM terminals (*Figure 4E*). As the behavioral traits caused by the downregulation of GABA-B-R3 are characteristic in optimism (*Carver et al., 2010*; *Solvi et al., 2016*), presynaptic control of reward signals may underlie such a cognitive bias. It would be fruitful to examine if a similar subcellular modulation of punishment-mediating neurons conversely leads to the pessimistic bias (*Sharot et al., 2009*; *Bateson et al., 2011*; *Sharot et al., 2012*; *Kregiel et al., 2016*; *Solvi et al., 2016*; *Zidar et al., 2018*).

## Materials and methods

**Key resources table**

| Reagent type (species) or resource | Designation | Source or reference | Identifiers | Additional information |
|---|---|---|---|---|
| Genetic reagent (*Drosophila melanogaster*) | GMR58E02-GAL4 | Bloomington *Drosophila* Stock Center | BDSC: 41347; FLYB: FBal0253714; RRID:BDSC_41347 | |
| Genetic reagent (*Drosophila melanogaster*) | GMR15A04-GAL4 | Bloomington *Drosophila* Stock Center | BDSC: 48671; FLYB: FBtp0057752; RRID:BDSC_48671 | |
| Genetic reagent (*Drosophila melanogaster*) | VT043924-GAL4 | Vienna *Drosophila* Resource Center | VDRC: v201194; FLYB: FBtp0105273; | |
| Genetic reagent (*Drosophila melanogaster*) | UAS-GABA-B-R3-RNAi | Bloomington *Drosophila* Stock Center | BDSC: 50622; FLYB: FBti0157477; RRID:BDSC_50622 | FlyBase symbol: P{TRiP.HMC02989}attP40 |
| Genetic reagent (*Drosophila melanogaster*) | UAS-mCD8::GFP | Bloomington *Drosophila* Stock Center | BDSC: 32194; FLYB: FBti0131936; RRID:BDSC_32194 | FlyBase symbol: P{20XUAS-IVS-mCD8::GFP} |
| Genetic reagent (*Drosophila melanogaster*) | UAS-GCaMP6s | Bloomington *Drosophila* Stock Center | BDSC: 42746; FLYB: FBti0151344; RRID:BDSC_42746 | FlyBase symbol: P{20XUAS-IVS-GCaMP6s} |
| Genetic reagent (*Drosophila melanogaster*) | GABA-B-R3-sfGFP-TVPTBF | Bloomington *Drosophila* Stock Center | VDRC: v318614; FLYB: FBst0491638; | FlyBase symbol: PBac{fTRG00613.sfGFP-TVPTBF} |
| Genetic reagent (*Drosophila melanogaster*) | UAS-GCaMP6s.brpS.mCherry | Bloomington *Drosophila* Stock Center | BDSC: 77131; FLYB: FBtp0125966; RRID:BDSC_77131 | |
| Antibody | (Rabbit polyclonal) anti-GFP | Invitrogen | Cat# A11122; RRID:AB_221569 | (1:1000) |
| Antibody | (Mouse monoclonal) anti-TH | ImmunoStar | Cat# 22941; RRID:AB_1267100 | (1:100) |
| Antibody | (Rabbit polyclonal) anti-DsRed | Clontech | Cat# 632496; RRID:AB_10013483 | (1:200) |
| Antibody | (Mouse monoclonal) anti-Brp | Developmental Studies Hybridoma Bank | Cat# nc82; RRID:AB_2314866 | (1:20) |

*Continued on next page*

*Continued*

| Reagent type (species) or resource | Designation | Source or reference | Identifiers | Additional information |
|---|---|---|---|---|
| Software, algorithm | GraphPad Prism 5 | GraphPad Software, San Diego, CA, 2007 | https://www.graphpad.com/ | |
| Software, algorithm | Fiji | MPI-CBG | https://fiji.sc/ | |

## Flies

Canton-S was used as a wild-type strain. *w;;R58E02-GAL4*, *w;R58E02-LexA*, and *w;;R15A04-GAL4* (*Jenett et al., 2012*; *Liu et al., 2012*), *yv;UAS-Grd-RNAi* (#58175), *yv;UAS-Lcch3-RNAi* (#50668), *yv; UAS-Rdl-RNAi* (#52903), *yv;UAS-GABA-B-R1-RNAi* (#51817), *yv;UAS-GABA-B-R2-RNAi* (#50608), *yv; UAS-GABA-B-R3-RNAi* (#50622), *UAS-Gad1-RNAi* (#51794), and *UAS-VGAT-RNAi* (#41958) (*Ni et al., 2011*), *yw;GABA-B-R3-T2A-GAL4* (#78976) (*Lee et al., 2018*), *w;;UAS-mCD8::GFP* (#32194) (*Pfeiffer et al., 2010*), *Tub-GAL80^{ts}* (*McGuire et al., 2003*), *w;UAS-GCaMP6s* (#42746) (*Chen et al., 2013*), *w;;LexAop-rCD2::GFP* (*Miyamoto et al., 2012*), *w;;VT043924-GAL4* (v201194) (*Wu et al., 2013*), *GABA-B-R3-sfGFP-TVPTBF* (v318614) (*Sarov et al., 2016*), and *UAS-GCaMP6s. brpS.mCherry* (#77131) (*Kiragasi et al., 2017*) were previously described. *w;UAS-mCD8::RFP* is a gift from Dr Ilona Kadow. Flies were raised at 24°C with 12:12 LD cycle. Knock-down flies were prepared as the F1 progeny of the crosses between females of *w;;R58E02-GAL4* or *w;UAS-Tub-GAL80^{ts};R58E02-GAL4* or *w* and males of UAS effectors or CS (*Figures 1A–B, E* and *4A–B* and *Figure 1—figure supplement 1A–B*), or females of UAS effectors or *w* and males of *w;;VT043924-GAL4* or *w* (*Figures 2E* and *4C–D*). The F1 progeny was raised at 24°C, aged to 6–12 days after eclosion before experiments. For the adult-specific knock-down experiment (*Figure 1B*), the F1 progeny was raised at 18°C and aged to 9–14 days after eclosion. For immunohistochemistry, a female reporter strain *w;;UAS-mCD8::GFP* (*Figure 1C–D*) or *w;UAS-mCD8::RFP;LexAop-rCD2::GFP* (*Figure 2C–D*) or *w;UAS-mCD8::RFP;R58E02-GAL4* (*Figure 2B*) or *UAS-GCaMP6s.brpS.mCherry* (*Figure 3A* and *Figure 3—figure supplement 1A*) was crossed to male GAL4 drivers or a reporter strain, *yw;GABA-B-R3-T2A-GAL4*, *w;R58E02-LexA;VT043924-GAL4*, *GABA-B-R3-sfGFP-TVPTBF*, or *w;;R15A04-GAL4*. Flies used for whole-mount immunohistochemistry were aged to 3–10 days after eclosion. For calcium imaging experiments, males of *w;UAS-GCaMP6s,UAS-mCD8::RFP* (*Figures 1F–H* and *2F–H*) or *UAS-GCaMP6s.brpS.mCherry* (*Figure 3B–G*) were crossed to *w;; R15A04-GAL4* or *w;UAS-GABA-B-R3-RNAi;R15A04-GAL4* females and raised at 24°C, aged to 3–8 days after eclosion, typically. For detailed fly genotypes used for experiments, see *supplementary file 1*.

## Behavioral assays

The conditioning and testing protocols were as described previously (*Yamagata et al., 2015*; *Yamagata et al., 2016*). Briefly, for a normal sugar learning experiment (*Figures 1A–B* and *2E* and *Figure 1—figure supplement 1A–B*), a group of approximately 50 flies in a training tube alternately received octan-3-ol (3OCT; Merck) and 4-methylcyclohexanol (4MCH; Sigma-Aldrich) for 1 min in a constant air stream with or without dried 2 M sucrose paper. For varied training duration protocol (*Figure 1E* and *Figure 1—figure supplement 1D*), flies received two odors and dried sugar alternately for defined duration (10–120 s) with an interval of 1 min between two odors. For odor generalization protocol (*Figure 4A–D*), flies were trained with an odor, that is, they alternately received 4MCH and paraffin oil (Sigma-Aldrich), or 3OCT and paraffin oil, for 1 min in a constant air stream with or without dried 2 M sucrose paper. Then the conditioned response of the trained flies was measured. For the normal protocol (*Figures 1A–B, E* and *2E*, *Figure 1—figure supplement 1A-B, D*), flies were given a choice between CS+ and CS- for 2 min in a T maze. For generalization protocol (*Figure 4A–D*), flies were given a choice between a 'trained' odor with a respective mixture ratio of a contaminant odor (2-methylcyclohexanol [2MCH]; Sigma-Aldrich or 1-octen-3-ol [1OCT]; Sigma-Aldrich) and the solvent for 2 min in a T maze. All odors were diluted to 10% in the paraffin

oil and placed in a cup with a diameter of 3 mm (OCT) or 5 mm (MCH). The memories were tested immediately after training unless otherwise stated. A learning index was then calculated by taking the mean preference of the two reciprocally trained groups. A half of the trained groups received reinforcement together with the first presented odor, and the other half with the second odor to cancel the effect of the order of reinforcement.

## Brain dissection, immunohistochemistry, and sample mounting

Dissection of fly brains was performed as previously described (*Kondo et al., 2020*) with minor modifications. Brains of female (*Figures 1C–D*, *2B–D* and *3A*, *Figure 2—figure supplement 1A–B* and *Figure 3—figure supplement 1A*) flies were dissected in PBS, pre-fixed in 1% paraformaldehyde (PFA) in PBS on ice up to 30 min, then fixed in 2% PFA in PBS for 1 hr at room temperature. Fixed brains were washed in PBT (0.1% Triton X-100 in PBS) for 3 × 10 min. Immunostaining was performed as previously described (*Kondo et al., 2020*). The following primary antibodies were used at the indicated dilution: rabbit anti-GFP (1:1000; Invitrogen; A11122), mouse anti-TH (1:100; ImmunoStar Inc; 22941), rabbit anti-DsRed (1:200; Clontech, 632496), or mouse anti-Brp (1:20; DSHB; nc82). The following secondary antibodies were used at the indicated dilution: AlexaFluor-488 goat anti-rabbit (1:1000; Invitrogen; A11034), Cy3 goat anti-rabbit (1:200; Jackson Labs), AlexaFluor-568 goat anti-mouse (1:1000; Invitrogen; 11004), AlexaFluor-568 goat anti-rabbit (1:250; Invitrogen; A11036), AlexaFluor-633 goat anti-mouse (1:200; Invitrogen; A21052). In *Figure 1C and D* and *Figure 2—figure supplements 1B*, 86% glycerol was used as a mounting medium, and the native GFP fluorescence was imaged without immunohistochemistry. In *Figures 2B–D* and *3A*, *Figure 2—figure supplement 1A* and *Figure 3—figure supplement 1A*, SeeDB2 (*Ke et al., 2016*) was used as mounting medium and either native or immunostained fluorescence was imaged.

## Confocal imaging

Imaging was performed on the Olympus FV1200 confocal microscope with GaAsP sensors. A 30×/1.05 silicone immersion objective (UPLSAPO30XS, Olympus) (*Figure 1C–D* and *Figure 2—figure supplement 1B*), or a 60×/1.42 oil immersion objective (PLAPON60XO, Olympus) (*Figures 2B–D* and *3A*, *Figure 2—figure supplement 1A* and *Figure 3—figure supplement 1A*) was used for scanning specific regions of interest (ROIs). A final voxel size of the image was $0.17 \times 0.17 \times 0.76$ µm$^3$ (*Figure 1C–D*), $0.11 \times 0.11 \times 0.45$ µm$^3$ (*Figure 2B* and *Figure 2—figure supplement 1A*), $0.21 \times 0.21 \times 0.43$ µm$^3$ (*Figure 2C–D*), $0.10 \times 0.10 \times 0.45$ µm$^3$ (*Figure 3A*), $0.51 \times 0.51 \times 0.68$ µm$^3$ (*Figure 2—figure supplement 1B*), and $0.79 \times 0.79 \times 0.37$ µm$^3$ (*Figure 3—figure supplement 1A*), respectively. Confocal stacks were analyzed with the open-source software ImageJ (National Institute of Health) and Fiji (*Schindelin et al., 2012*). Where appropriate, 2D/3D image deconvolution was applied using Diffraction PSF 3D and Parallel Iterative Deconvolution plugins in ImageJ.

## Fly preparation and in vivo calcium imaging

Flies were treated as described in *Hiroi et al., 2013*; *Shiozaki and Kazama, 2017* with some modifications. The fly was briefly (<1 min) anesthetized on ice and placed in a custom-made holding device on a Peltier plate (CP-085, Scinics) held at 4°C. The head capsule was fixed to the dish by UV curing optical adhesives (NOA68, Thorlabs). The proboscis was glued onto the capsule to eliminate brain movement. Forelegs interfering sugar feeding during recordings were removed. A small window on the top of the head capsule was opened using sharp forceps in *Drosophila* saline (103 mM NaCl [31320–05, Nacalai tesque], 3 mM KCl [28514–75, Nacalai tesque], 5 mM TES [32810–55, Nacalai tesque], 8 mM Trehalose [Tokyo kasei kogyo], 10 mM D-glucose [16806–25, Nacalai tesque], 26 mM NaHCO$_3$ [31213–15, Nacalai tesque], 1 mM NaH$_2$PO$_4$ [A0110846 010, Merch], 1.5 mM CaCl$_2$ [C5080, Sigma-Aldrich], 4 mM MgCl$_2$ [M2670, Sigma-Aldrich], ~270 mol/kg, pH ~7.2). Air sacs and fat bodies covering the brain surface were carefully removed.

A laser scanning confocal microscope (A1R, Nikon) equipped with a 30×/1.1 water immersion objective (Apo LWD 25×, Nikon) and a Piezo nanopositioner (Nano-F450, MCL Inc) combined with a Nano-Driveone controller (MCL Inc) was used for live imaging. GCaMP6s and mCD8::RFP or mCherry were sequentially excited at 488 and 561 nm, respectively. The emission light was collected onto GaAsP detectors using dichroic mirrors and emission filters (BP500–550 and BP570-620). Transverse sections of the MB lobes and the superior medial protocerebrum were scanned at a resolution

of 0.5 μm/pixel (512 × 128 pixels) at 333 ms/frame (*Figure 1F–H*), 0.5 μm/pixel (512 × 256 pixels) at 1 s/frame (*Figure 2G–H* and *Figure 2—figure supplement 1C*), or 0.33 μm/pixel (512 × 128 pixels) at 1 s/frame (*Figure 3B–E* and *Figure 3—figure supplement 1A–D*) with line scans with 4× (*Figures 1F–H* and *2G–H* and *Figure 2—figure supplement 1C*) or 16× (*Figure 3B–E* and *Figure 3—figure supplement 1*) averages using the resonant scanning mode. The pinhole was set to 2.5 AU (561 nm). For 3D imaging, two z sections (ca. 100 μm interval) were scanned using a piezo-electric motor. To record calcium responses to sugar and odors, images were acquired for 20 or 30 s and saved for later image processing. For sugar stimulation, a droplet of 500 mM sucrose deposited on a tip of Microloader pipette tip (Eppendorf) was presented to the proboscis for 3 s using a micro-manipulator (UN-3C, Narishige). Sugar stimulation to flies was monitored with a USB camera (Grass-hopper3, FLIR) mounted with a zoom lens (MACRO ZOOM 0.3×−1 × 1:4.5, Computar) and captured by FlyCapture2 (FLIR). Odor stimulation has been made manually using a 50 ml syringe containing a piece of filter paper (1 × 2 cm$^2$) soaked with pure or 10 times diluted 4MCH and 3OCT. For each stimulation, ~15 ml odor contained air was delivered to a fly in 3 s through a 4 mm silicon tube placed ca. 10 mm away from the fly head.

## Data analyses

All the acquired images were first processed with Fiji. An object in each recording was stabilized by TurboReg plugin (*Thévenaz et al., 1998*) using mCD8::RFP or mCherry signal. ROIs to be involved in later calculations were defined by mCD8::RFP signal in the left or right hemisphere. In *Figures 1F–H* and *2F–H* and *Figure 2—figure supplement 1C*, GCaMP6s signal was used as a fluorescent F value. The ΔF/F$_0$ was calculated as:

$$\Delta F/F_0 = \frac{F_t - F_0}{F_0},$$

where $F_t$ and $F_0$ denote fluorescent values at time frame t and baseline (i.e., ~7 frames before stimulation), respectively. To highlight MB compartments that responded to stimulations (*Figure 1F*), a time series projection of the ΔF/F$_0$ during stimulation (for 3 s) was thresholded and superimposed on a projection image of mCD8::RFP signal at respective frames.

After XY registration, the Brp::GCaMP6s signal was divided by mCherry (GCaMP6s/mCherry) to normalize the calcium signal by Bruchpilot abundance. An ROI for the α1 compartment of the MB was defined by mCherry signal. Pixels devoid of an mCherry fluorescence value were censored. The image stacks were then imported to Matlab (MathWorks) and log-transformed.

To evaluate the spatial pattern of the calcium intensity in the PAM-α1 terminals, we computed the size of the area of each peak (*Figure 3F*) in the following manner: (i) For each pixel in an image, the pixel with the largest intensity value among the four neighboring pixels was identified. If the given pixel had a larger intensity value than those of the neighboring pixels, we recorded the pixel as a local peak. We ignored pixels that had the background intensity value. This procedure yielded the steepest path to a local peak from each pixel. (ii) We clustered the pixels based on the local peak connected by the paths. Note that this simple algorithm is a discrete version of the gradient ascent algorithm and used for clustering with height information elsewhere (*Ezaki et al., 2017*). (iii) We counted the number of pixels belonging to the same cluster and computed the average cluster size for each fly based on nine frames before stimulation (or three frames during the stimulation). Because the average cluster size may be affected by the size and shape of the PAM-α1 terminals in the recorded images, we normalized it by a null model. The average cluster size for the null model was obtained as follows. (iv) For each image, we shuffled the intensity values across the pixels. (v) We computed the average cluster size for the shuffled image by performing (i)–(iii). (vi) We repeated (iv) and (v) for 1000 runs. (vii) Finally, we calculated the average cluster size for the null model by averaging the results over the 1000 runs and nine frames before stimulation (or three frames during the stimulation). This value was used for normalization in *Figure 3F*.

In addition to the cluster size analysis above, we used another measure (i.e., peak size) to quantify the spatial structure of the calcium intensity. The peak size was computed for each image as follows. First, we collected the pixels that had an intensity value larger than 95 percentile of the entire pixels in each image. Then, we identified the clusters of these selected pixels by checking if multiple pixels (peaks) were adjacent to each other at the top, bottom, left, or right. Finally, we counted the

number of pixels in each cluster and computed the average. Similarly to the cluster size measure above, we normalized this value by using a null model. We computed the peak size for 1000 randomized images that were obtained by shuffling the intensity values in the pixels in the original image. The results were averaged over the 1000 null data, which we used for the normalization.

## Statistics

Statistics were performed by Eclipse (Eclipse foundation) and Prism5 (Graphpad). For the data points that did not violate the assumption of normality and homogeneity of variance (D'Agostino and Brown-Forsythe test), parametric statistics were applied. The data points that were significantly different from the normal distribution were analyzed with nonparametric statistics. The significance level of statistical tests was set to 0.05. For details, see *Supplementary file 1*.

To estimate the acquisition curve dynamics (*Figure 1E*), hyperbola curve fitting was applied:

$$\text{LI} = \frac{\text{A}t}{\text{B} + t}$$

where A and B are constants, t is the training duration, and LI is learning index. 'A' denotes the theoretical maximum value of LI (i.e., plateau) and 'B' the training duration required to reach the half of the maximum (i.e., acquisition speed). To test the statistical significance of observed differences in A and B ($\Delta A_{obs.}$ and $\Delta B_{obs.}$) between genotypes, we performed permutation tests (*Knijnenburg et al., 2009*); we randomly shuffled the experimental dataset by reassigning the group labels and fitted a hyperbola function to the data to calculate the differences in A and B between groups ($\Delta A_{perm.}$ and $\Delta B_{perm.}$). The procedure was repeated over 2000 runs to generate the null distributions of $\Delta A_{perm.}$ and $\Delta B_{perm.}$ for testing the statistical significance of $\Delta A_{obs.}$ and $\Delta B_{obs.}$.

## Acknowledgements

We thank I Kadow, V Ruta, BDSC and VDRC for sharing fly stocks; T Hige, M Hiroi, H Kazama, HM Shiozaki, GC Turner for our setting up calcium imaging experiments; T Ichinose for helping statistics; Y Aso, S Kondo, M Tanaka, K Hamaguchi, Y Hirano for stimulating discussions and for commenting on the manuscript.

# Additional information

## Funding

| Funder | Grant reference number | Author |
|---|---|---|
| Japan Society for the Promotion of Science | 19KK0383 | Nobuhiro Yamagata |
| Japan Society for the Promotion of Science | 17H04765 | Nobuhiro Yamagata |
| Japan Society for the Promotion of Science | 20H05525 | Hiromu Tanimoto |
| Japan Society for the Promotion of Science | 19K22577 | Hiromu Tanimoto |
| Japan Society for the Promotion of Science | 17H01378 | Hiromu Tanimoto |

The funders had no role in study design, data collection and interpretation, or the decision to submit the work for publication.

## Author contributions

Nobuhiro Yamagata, Conceptualization, Data curation, Formal analysis, Supervision, Funding acquisition, Validation, Investigation, Visualization, Methodology, Writing - original draft, Project administration, Writing - review and editing; Takahiro Ezaki, Data curation, Software, Formal analysis, Visualization, Methodology, Writing - original draft; Takahiro Takahashi, Hongyang Wu,

Investigation; Hiromu Tanimoto, Conceptualization, Data curation, Formal analysis, Supervision, Funding acquisition, Validation, Writing - original draft, Project administration, Writing - review and editing

### Author ORCIDs
Nobuhiro Yamagata (iD) https://orcid.org/0000-0003-1993-2038
Takahiro Ezaki (iD) https://orcid.org/0000-0003-4175-3028
Takahiro Takahashi (iD) https://orcid.org/0000-0002-3584-7565
Hongyang Wu (iD) http://orcid.org/0000-0001-7889-6941
Hiromu Tanimoto (iD) https://orcid.org/0000-0001-5880-6064

### Decision letter and Author response
Decision letter https://doi.org/10.7554/eLife.64907.sa1
Author response https://doi.org/10.7554/eLife.64907.sa2

## Additional files

### Supplementary files
• Supplementary file 1. List of fly strains and crosses for experiments, and the statistical results.

• Transparent reporting form

### Data availability
All data is available in the main text or the supplementary materials, or in the following data repositories (Source Data: https://gin.g-node.org/Nobu/Yamagata_Data_Source, Source Code: https://github.com/tkEzaki/peak_size_analysis; copy archived at https://archive.softwareheritage.org/swh:1:rev:ba29c4fb5f303958286f1a2f90fe8a8b7cb3ea84).

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
