## [Decision Letter]

**Acceptance summary:**

Learning in this system is based on dopamine-induced synaptic plasticity. The present work demonstrates that the dopaminergic inputs can be modulated by pre-synaptic inhibition from a GABAergic neuron in the circuit. So this is one way the circuit can control the strength of the reward signal reaching the learning centers, even when the sensory inputs remain the same. It adds a new level of complexity to reward signaling pathways in the brain,

**Decision letter after peer review:**

Thank you for submitting your article "Presynaptic inhibition of dopamine neurons controls optimistic bias" for consideration by *eLife*. Your article has been reviewed by 2 peer reviewers, and the evaluation has been overseen by K VijayRaghavan as the Senior and Reviewing Editor. The reviewers have opted to remain anonymous.

This work studies how *Drosophila* mushroom body dopaminergic- neurons are affected by inhibitory inputs. Learning in this system is based on dopamine-induced synaptic plasticity. The present work demonstrates that the dopaminergic inputs can be modulated by presynaptic inhibition from a GABAergic neuron in the circuit, demonstrating one way the circuit can control the strength of the reward signal reaching learning centers, even when the sensory inputs remain the same. The study adds a new level of complexity to reward signaling pathways in the brain.

There are substantial concerns that the authors are requested to address before acceptance. These are delineated in the consolidated reviews below.

Essential revisions:

1. Figure 1B is an important basis of the whole work. Considering that the temperature may significantly affect appetitive learning performance, the parental control groups at 18℃ should be completed.

2. Since the RNAi strategy might have an off-target problem, checking the efficiency of GABA-B-R3-RNAi using Q-PCR or employing another independent RNAi line of GABA-B-R3 is helpful.

3. Considerable attention needs to be given to the calcium imaging of DAN active zones in Figure 3. What, in the authors' view, is the biological significance of a larger cluster size? It is also not clear how that increase connects to the behavioral results about generalization. Given that these patterns are odor-evoked rather than reward-evoked, how can we understand their role in learning-related plasticity? They need to be clarified since on p10, lines 8-10, they write: 'These results suggest that GABA-B-R3 function at the presynaptic terminal is necessary for the fine spatial regulation of the reward signal'. However, their experiments do not examine the reward signal itself. Further, the authors need to clarify whether they think the increase in cluster size is simply a reflection of greater amplitude activity in these cells, or whether new 'active zones' are recruited when inhibition is blocked. Would they see the same increase simply using higher odor concentrations? Simply increasing DAN activity could potentially have the same effect on memory as 'blurring DAN activity', but the authors are not clear about the distinction in the text.

4. There are important technical issues with specific panels in Figure 3.

Figure 3A: There is no validation that these spots are individual active zones. There should be some co-localization with other presynaptic proteins to even begin this direction. And there is almost certainly not enough resolution to say these are individual AZs (axial resolution is at least 4 microns so there are almost certainly many AZs collapsed on one another in their images). This could be addressed with a more measured language in the text (i.e. don't write 'we measured calcium influx at individual active zones' p.5 line 20-21).

Figure 3B: How would GABAR knockdown broaden the distribution on the low end (left side of x-axis)? We can see how it would increase the mean intensity, but it seems that both ends of the distribution expand in this analysis. How robust is the effect? Is it seen with a different set of 9 frames? Why only analyze 9 frames, why not do all frames?

Figure 3D is a confusing way of representing imaging data. We would prefer to see a video or a per pixel dF/F heat map of what the stimulus responses looked like. I also don't see the biological significance of the diminished inverse correlation between mean and SD in the GABAR knockdown.

Figure 3E and F: What are the cluster sizes in microns rather than arbitrary units? In Fig3F the difference in cluster sizes +-GABAR RNAi is very small. If we go from how they calculated what 'normalized cluster size' should be, then the difference between the null distribution of sizes and their observations in WT (1.055 -1) is less than the difference +- GABAR RNAi (1.065-1.055). So is this really a significant difference? Does it make sense when expressed in physical distance?

How different are dFF values between clusters? The graphic in Fig3E is color-coded by cluster size (in pixels), not by response amplitude.

Overall it would be much more informative to simply show a zoom of the ROI displaying the dFF values they observed.

5. A general technical comment, which should be addressed: Will not increasing response amplitude naturally lead to an increase in cluster sizes, as there is more signal to poke out above the background? If the authors looked at higher odor intensities, would they also see an increase in cluster size?

Important points on the interpretation of experiments:

6. Importantly, the experiments in Figure 3 don't look at DAN responses to reward (sucrose), they look at odor responses. It is not clear what that has to do with plasticity. It will be better to look at responses to sucrose for this whole question.

Based on the text on p.10 lines 1-3 it sounds like the authors think the calcium they measure in the DAN terminals carries a representation of the odor i.e. is different for distinct odors. And the increase in cluster size is presumed to decrease the complexity of the pattern, and again presumably make patterns for different odors less distinct. However, according to p.16 line 20, these clusters are derived from analysis of 9 frames *prior to stimulation*. So how could they carry an odor representation (as claimed p.10 lines 1-3) if the odor has not yet been presented?

7. How consistent are these clusters? Should we be shown results from repeated odor presentations, but maybe that's not the claim? But shouldn't they at least be consistent across timepoints?

8. There is no metric for the complexity of the patterns, only the size of the clusters. How odor-specific are these patterns (how distinct for the odors in question, and how consistent across repeated presentations)?

9. The most likely explanation for their results seems to be that the reward signal gets stronger and broader (akin to increasing the concentration of sucrose). But since they're imaging odor responses here, we are not sure what they're trying to say, the text describing their overall interpretation (p.10) is not easily digestible.

10. In Figure 4: Since GABAR RNAi flies start with a higher learning score, the authors should compensate for this when evaluating whether they generalize more than controls. Generalization is evident in Figure 4A, but in 4B the red line seems to be just an upward shift of the other lines. The difference at 50:50 is perhaps not increased generalization, rather it is just a consequence of increased overall learning.

---

## [Author Response]

Essential revisions:1. Figure 1B is an important basis of the whole work. Considering that the temperature may significantly affect appetitive learning performance, the parental control groups at 18°C should be completed.

We agree with the reviewer’s point that temperature may significantly affect flies’ performance. We thus conducted a control learning experiment using the same genotypes raised at 18℃ to permit the GAL80[ts] suppression of transgenic RNAi. No statistical differences were observed across genotypes, suggesting a physiological rather than a developmental effect of GABA-B-R3-KD on performance enhancement. This new data is now incorporated into Figure 1B.

2. Since the RNAi strategy might have an off-target problem, checking the efficiency of GABA-B-R3-RNAi using Q-PCR or employing another independent RNAi line of GABA-B-R3 is helpful.

The reviewer might have overlooked them, but we did indeed include consistent behavioral data using the ‘second’ RNAi strain in our original manuscript. They are presented in Figure 1—figure supplement 1B.

3. Considerable attention needs to be given to the calcium imaging of DAN active zones in Figure 3. What, in the authors' view, is the biological significance of a larger cluster size? It is also not clear how that increase connects to the behavioral results about generalization. Given that these patterns are odor-evoked rather than reward-evoked, how can we understand their role in learning-related plasticity? They need to be clarified since on p10, lines 8-10, they write: 'These results suggest that GABA-B-R3 function at the presynaptic terminal is necessary for the fine spatial regulation of the reward signal'. However, their experiments do not examine the reward signal itself. Further, the authors need to clarify whether they think the increase in cluster size is simply a reflection of greater amplitude activity in these cells, or whether new 'active zones' are recruited when inhibition is blocked. Would they see the same increase simply using higher odor concentrations? Simply increasing DAN activity could potentially have the same effect on memory as 'blurring DAN activity', but the authors are not clear about the distinction in the text.

We thank the reviewer for raising these important potential problems, as these explanations were indeed unclear. We made a major textual revision for Figure 3 to better explain why we characterize active-zone calcium patterns in the PAM terminals. Now it extends to 4 paragraphs (P5, L119-P12, L256).

The reviewer also raised an interesting question regarding the interpretation of increased cluster size (Figure 3F). We made a scatter plot to analyze the relationship between peak size and calcium signal intensity (Author response image 1). This revealed no clear correlation between them. Furthermore, a similar plot for GABA-B-R3 knock-down terminals, which responded to odor stimulation with larger calcium increase, did not show a greater correlation (even slightly smaller dependency on the contrary [slope of the regression: 0.0543 vs. 0.0517 in WT and GABA-B-R3-KD flies, respectively]). This suggests that the cluster size is controlled by other mechanism(s) than the amplitude (see also Author response image 2). One such possibility would be the recruitment of adjacent active zones. We now added this information in the revised text (P12. L247- P12. L249)

**Author response image 1. sa2fig1:** Scatter plots showing dependency between the size (abscissa) and intensity (ordinate) of each peak calculated by the method we described. Each circle represents a single cluster in the 9 frames before stimulation of a single representative fly.

**Author response image 2. sa2fig2:** The cluster value map. Each cluster is color-coded by the averaged intensity of associated pixels. Left: control, right: GABA-B-R3-KD.

4. There are important technical issues with specific panels in Figure 3.Figure 3A: There is no validation that these spots are individual active zones. There should be some co-localization with other presynaptic proteins to even begin this direction. And there is almost certainly not enough resolution to say these are individual AZs (axial resolution is at least 4 microns so there are almost certainly many AZs collapsed on one another in their images). This could be addressed with a more measured language in the text (i.e. don't write 'we measured calcium influx at individual active zones' p.5 line 20-21).

Following the suggestion, we revised the corresponding text by providing a more factual result description (P5, L120, P8, L163-165).

We validated the localization of the sensor to the active zones in the PAM terminals. The calcium sensor is fused to *brp-short* (Kiragasi et al., Cell Rep 2017), a widely accepted molecular tag for the localization in the active zone. While the monoclonal antibody against Brp (nc82) does not recognize the Brp^short^ in the sensor (Schumid et al., Nat Neurosci, 2008), Brp^short^::GCaMP6s::mCherry expressed in the PAM neurons nicely colocalized with nc82 signal, indicating sensor integration into native active zones. This proper sensor integration is consistent with the observation in the motor neurons (Kiragasi et al., Cell Rep 2017). We now present this new data in the revision (P8, L165-167, Figure 3—figure supplement 1A).

Figure 3B: How would GABAR knockdown broaden the distribution on the low end (left side of x-axis)? We can see how it would increase the mean intensity, but it seems that both ends of the distribution expand in this analysis. How robust is the effect? Is it seen with a different set of 9 frames? Why only analyze 9 frames, why not do all frames?

Since the intensity distributions in Figure 3B are representative examples, it does not make too much sense to discuss the broadened tails in this knock-down fly. The intention and the point of these Figure 3B and 3C are to show that GABA-B-R3 is required for the regulation of terminal calcium levels. We now mention the tendency of broader distributions of the Brp::GCaMP6s::mCherry signals upon knock down, in addition to increased temporal fluctuations, suggesting compromised robustness. In addition, we provide textual explanations regarding these analyses (P8. L173-180).

Figure 3D is a confusing way of representing imaging data. We would prefer to see a video or a per pixel dF/F heat map of what the stimulus responses looked like. I also don't see the biological significance of the diminished inverse correlation between mean and SD in the GABAR knockdown.

Since the mean and SD are the primary statistics that represent the Gaussian distribution, we mapped our data in the logarithm mean vs. sd space (Figure 3E). Although the biological importance of the observed inverse correlation is unclear, it suggests that each individual has a unique ‘set point’ of activity level and heterogeneity in the dopamine terminals, and such a control requires GABA-B-R3. The structure of these set points among individuals might be relevant to the “criticality” state of a system for efficient information processing (e.g., Hesse and Gross, 2014; Ezaki et al., Communications biology 2020). We mentioned this in the figure legend (P9, L194-196, P10, L202-205).

Given sample motion during measurement, pixel-wise dF/F would not be a powerful approach. To be a little more reader friendly, we added separate mean and sd traces in the main figure of the revised manuscript (Figure 3D). The original files for our analyses are also deposited online (https://gin.g-node.org/Nobu/Yamagata_Data_Source).

We would like to remind that one of the significant advantages of using the Brp::GCaMP6s::mCherry sensor is that each molecule has the mCherry reference. This makes the measurement more robust against motion artifacts and sensor expression levels (P8. L163-165).

Figure 3E and F: What are the cluster sizes in microns rather than arbitrary units? In Fig3F the difference in cluster sizes +-GABAR RNAi is very small. If we go from how they calculated what 'normalized cluster size' should be, then the difference between the null distribution of sizes and their observations in WT (1.055 -1) is less than the difference +- GABAR RNAi (1.065-1.055). So is this really a significant difference? Does it make sense when expressed in physical distance?

The lateral resolution of our images is 0.33 um/px, yielding physical cluster sizes of 0.536 and 0.544um^2^ for the control and GABA-B-R3 KD samples respectively. Since we normalized these values with the mean cluster size of the randomly reshuffled images, these values do not have a unit.

To further support this cluster size analysis, we examined another metrics, i.e., peak size, which quantifies the size of bright pixels (brighter than the 95 percentile of the entire pixel intensities in a given ROI). This metrics is thus independent of overall intensities but also quantifies the fineness of the pattern. This measure also supported our conclusion: GABA-B-R3 KD caused larger peak sizes. This new data is now incorporated into the text (P12 L249-254), and supplementary figure (Figure 3—figure supplement 1F).

How different are dFF values between clusters? The graphic in Fig3E is color-coded by cluster size (in pixels), not by response amplitude.

We mapped the calcium intensity of each cluster, but did not find a clear correlation with the cluster size (Author response image 2). As shown in Author response image 1, the amplitude does not well explain the cluster size.

Overall it would be much more informative to simply show a zoom of the ROI displaying the dFF values they observed.

For the reasons we already argued in our response to the comment 4 above, we mainly focus on the frame-wise spatial pattern of active-zone calcium activities.

5. A general technical comment, which should be addressed: Will not increasing response amplitude naturally lead to an increase in cluster sizes, as there is more signal to poke out above the background? If the authors looked at higher odor intensities, would they also see an increase in cluster size?

This is essentially the same concern which we addressed above (Author response images 1 and 2).

Important points on the interpretation of experiments:6. Importantly, the experiments in Figure 3 don't look at DAN responses to reward (sucrose), they look at odor responses. It is not clear what that has to do with plasticity. It will be better to look at responses to sucrose for this whole question.

Basically, this is the repetition of the first part of Point 3 above. These explanations were indeed unclear. We made a major textual revision for Figure 3 to better explain why we characterize active-zone calcium patterns in the PAM terminals. Now it extends to 4 paragraphs (P5, L119-P12, L256).

Based on the text on p.10 lines 1-3 it sounds like the authors think the calcium they measure in the DAN terminals carries a representation of the odor i.e. is different for distinct odors. And the increase in cluster size is presumed to decrease the complexity of the pattern, and again presumably make patterns for different odors less distinct. However, according to p.16 line 20, these clusters are derived from analysis of 9 frames prior to stimulation. So how could they carry an odor representation (as claimed p.10 lines 1-3) if the odor has not yet been presented?

The reviewer correctly understands our points. Since the local active-zone calcium influx in the PAM terminals reflects KC odor activity, it should be related to evaluating odors.

As resting-state functional connectivity predicts how the circuit operates tasks (e.g., Cole et al., Nat Neurosci, 2016), we analyzed its basal activity in the PAM terminals. In response to the reviewer’s suggestion, we now analyzed the cluster size during odor stimulation. Indeed, we found the increased cluster size in the knock-down terminals, a consistent result with baseline activity (Figure 3G left). This new data is now presented in the main figure (Figure 3G right) and text (P12. L244-247).

7. How consistent are these clusters? Should we be shown results from repeated odor presentations, but maybe that's not the claim? But shouldn't they at least be consistent across timepoints?

Following the reviewer’s request, we calculated cluster sizes for all frames before and during odor stimulation (Author response image 3). The result clearly shows consistently greater cluster size in GABA-B-R3 knock-down flies over time.

**Author response image 3. sa2fig3:** Time courses of normalized cluster size (left) and the comparison between genotypes using all frames from pre- and during-stimulation phases (frames 1-12, right). The averaged cluster size was consistently higher in the GABA-B-R3 knock-down flies compared to wild-type flies. * P < 0.05, Mann Whitney test. N=15, 11.

8. There is no metric for the complexity of the patterns, only the size of the clusters. How odor-specific are these patterns (how distinct for the odors in question, and how consistent across repeated presentations)?

We meant the spatial ‘blurring’ as larger local cluster sizes, but replaced this word to avoid any unnecessary confusions by readers (P9, L190, P12, L255).

9. The most likely explanation for their results seems to be that the reward signal gets stronger and broader (akin to increasing the concentration of sucrose). But since they're imaging odor responses here, we are not sure what they're trying to say, the text describing their overall interpretation (p.10) is not easily digestible.

Thank you for the comment. Our model is that the sugar reward signal is globally inhibited in the PAM terminals by GABA-B-R3, and locally disinhibited by the input from odor-representing KCs to selectively enhance dopaminergic reward signals to these KCs. This has been clarified through the text revision (P12, L254-256), and revising a figure panel (Figure 4E).

10. In Figure 4: Since GABAR RNAi flies start with a higher learning score, the authors should compensate for this when evaluating whether they generalize more than controls. Generalization is evident in Figure 4A, but in 4B the red line seems to be just an upward shift of the other lines. The difference at 50:50 is perhaps not increased generalization, rather it is just a consequence of increased overall learning.

We understand the reviewer’s concern. To evaluate the generalization curve irrespective of the intensity, we normalized the data by dividing the average score of the 100:0 mixture (i.e., testing with the trained odor), and found essentially the same results (Author response image 4). Significant reduction of odor generalization at the 50:50 mixture was detected in the control groups, but not in the experimental group (Author response image 4). These results are thus consistent with our previous presentation, and indicate that GABA-B-R3 in the PAM terminals is required for the specificity of rewarded memory to the learned odor.

**Author response image 4. sa2fig4:** Normalized generalization curves using 4MCH:2MCH (left) and 3OCT:1OCT (right) mixtures. Statistics suggest experimental group exhibit altered memory specificity to learned odors when compared with its parental control groups. (4MCH:2MCH) R58E02-GAL4/+, *p* = 0.030; UAS-GBR3-RNAi/+, *p* = 0.006; R58E02-GAL4/UAS-GBR3-RNAi, *p* = 0.972. N = 8-17. (3OCT:1OCT) R58E02-GAL4/+, p = 0.002; UAS-GBR3-RNAi/+, p = 0.006; R58E02-GAL4/UAS-GBR3-RNAi, *p* = 0.500. N = 8-20. Tukey’s multiple comparisons test.

To corroborate the result with receptor knock down, we performed a new set of generalization experiments upon silencing GABA synthesis in the APL neurons. Strikingly, APL-GAL4/UAS-GAD1-RNAi flies showed significantly broad generalization profile, very similar to the results of GABA-B-R3 knock down in the PAM neurons (Figure R9). We confirmed this broadened generalization with two different odors (Figure R9). We now added these new data in the revised text (P12. L263-264; Figures 4C, 4D).